# The Cover Depth Effect on Corrosion-Induced Deterioration of Reinforced Concrete Focusing on Water Penetration: Field Survey and Laboratory Study

**DOI:** 10.3390/ma14133478

**Published:** 2021-06-22

**Authors:** Shingo Asamoto, Junya Sato, Shinichiro Okazaki, Pang-jo Chun, Raktipong Sahamitmongkol, Giang Hoang Nguyen

**Affiliations:** 1Department of Civil and Environmental Engineering, Saitama University, Saitama 338-8570, Japan; 2Consulting Headquarters, Yachiyo Engineering Co., Ltd., Tokyo 111-8648, Japan; jn-sato@yachiyo-eng.co.jp; 3Faculty of Engineering and Design, Kagawa University, Takamatsu 761-0396, Japan; okazaki.shinichiro@kagawa-u.ac.jp; 4Institute of Engineering Innovation, School of Engineering, The University of Tokyo, Tokyo 113-8656, Japan; chun@g.ecc.u-tokyo.ac.jp; 5Department of Civil Engineering, King Mongkut’s University of Technology Thonburi, Bangkok 10140, Thailand; Raktipong.sah@kmutt.ac.th; 6Inspection and Maintenance Division, National University of Civil Engineering, Hanoi 11616, Vietnam; giangnh@nuce.edu.vn

**Keywords:** reinforced concrete bridge, reinforcement corrosion, corrosion-induce deterioration, water penetration, cover depth, drying and wetting conditions

## Abstract

Reinforced concrete bridges were visually surveyed in Japan, Thailand, and Vietnam to study the deterioration caused by internal steel corrosion under different climates, focusing on the concrete cover depth. Spalling or cracking arising from corrosion is likely where water is supplied. According to prior studies and our surveys, a concrete cover depth of more than 40 mm was found to prevent spalling, regardless of environmental conditions and structure age. Because water supply at steel is a key corrosion factor, it was hypothesised that under natural conditions, the water penetration in concrete would remain at a depth of approximately 40 mm. Our laboratory study examined water penetration under drying and wetting conditions. The results also suggested that under periodic rainfall conditions, the threshold of water penetration was not exceeded. The numerical study indicated maximum moisture evaporation to facilitate oxygen diffusion occurred at a depth of approximately 30–40 mm unless the concrete was exposed to continuous drying for more than one month. It was experimentally and numerically concluded that an adequate cover depth of greater than 40 mm could inhibit moisture and oxygen penetration at the steel, which supported the survey findings of cover depth effect on a high resistance to corrosion-induced deterioration despite an increase in service life.

## 1. Introduction

Steel corrosion in reinforced concrete structures is critical to structural performance and also causes spalling of concrete cover, which poses a risk to occupants and any other person passing under the structure. It is well established that corrosion is initiated after the depassivation of the steel surface caused by the carbonation of the cover concrete and chloride ingress. In practical design, the risk of corrosion is generally assessed based on the carbonation rate and chloride diffusion to determine the design cover depth [1,2,3]. The supply of both water and oxygen is necessary for the initiation of corrosion after depassivation. Tuutti [4] investigated the effect of relative humidity (hereafter referred to as “RH”) during the propagation stage of corrosion, considering chloride ingress and diffusion of O_2_, and observed a lower corrosion rate in carbonated concrete with a decreased RH. Similarly, Gonzalez et al. [5] reported an environment with a low RH (50%) can limit the corrosion rate even when carbonation occurs and chloride ions are present in the mortar. Furthermore, Glass et al. [6] concluded that high RH and chloride presence significantly increased the risk of corrosion in carbonated mortars. Moreover, Stefanoni et al. [7] summarised the key factors affecting the corrosion rate of reinforcement in carbonated concrete and concluded that the most important parameter for steel corrosion is the exposure condition, which changes the pore saturation.

It has been classically known that carbonation is one of the keys to initiate corrosion. For real structures in the field, it has been reported that corrosion was unlikely to be observed when the concrete was directly kept from moisture exposure, even when carbonation reached the reinforcing steel; however, corrosion can occur because of rainfall moisture and other sources [8,9]. Ishibashi et al. [10] and Maehara and Iyoda [11] reported that moisture supply is more likely to cause spalling in reinforced concrete. In Japan, the effect of water penetration on corrosion has been considered [12,13], along with carbonation assessment in the design specifications to utilise blended cement, reported to have a higher carbonation rate than concrete without mineral admixtures [14,15,16]. The simple design water penetration rate determined by the water-to-binder ratio, which is proportional to the square root of time based on the Lucas–Washburn equation, is used to calculate the penetration, assuming short-term penetration in a practical way [12,13]. The design cover depth to prevent corrosion in service life is determined by taking into account the age when water penetrates the steel position, as well as the carbonation progress and chloride ingress.

It is known that the total water uptake in unsaturated concrete is initially proportional to the square root of the elapsed time, but it is gradually retarded [17], which is also observed in porous materials other than cementitious materials [18]. Recently, McDonald et al. [19] explained the anomalous water sorption in cement pastes by considering the dynamic porosity based on the 1H nuclear magnetic resonance results, and the proposed mechanism was verified by a transport model in three types of pores with a variable porosity fraction according to the water saturation. Previous studies have shown the effect of water infiltration on corrosion in reinforced concrete for both laboratory specimens and real structures; however, the infiltration mechanism in concrete is anomalous, leading to difficulty in determining the moisture behaviour in concrete, especially under outdoor drying and wetting conditions.

In reality, because the water penetration rate can be affected by climate, such as ambient temperature and precipitation, the effect of cover depth on the deterioration of reinforced concrete by corrosion with water penetration in tropical regions can be different from that reported in Japan. In addition, the water penetration is gradually retarded, as reported previously, which might prohibit water from penetrating deeply. Hence, the objective of this study is to examine and compare the effect of cover depth on the deterioration of reinforced bridges by steel corrosion under different climate environments in Japan, Thailand, and Vietnam. The concrete cover depth was measured using a non-destructive method focusing on the spalling or cracking caused by the corrosion of the internal reinforcement. Subsequently, one-dimensional water penetration in concrete with different cover depths under drying and wetting conditions was experimentally studied, and an experimental method was proposed to focus on the electivity change before and after wetting at each depth. The experimental results are also discussed based on the numerical analysis of the drying process in the capillary and gel pores. The significance of this study is to survey the cover depth effect on corrosion-induced deterioration on-site in different Asian countries focusing on the water penetration rather than the classical corrosion factors, such as carbonation and chloride ingress, and then to experimentally and numerically investigate the threshold of water penetration and drying in cover concrete to support the survey findings.

## 2. Field Surveys for Assessment of Corrosion and Cover Depth of Reinforced Concrete Bridges

### 2.1. Location and Inspection

To compare the deterioration characteristics and examine the effect of cover depth on the deterioration in different Asian countries, field surveys of reinforced concrete bridges were conducted in Japan, Thailand, and Vietnam. Visual inspection was carried out to observe cracking, flaking, spalling, and rust arising from corrosion, with a focus on the boundary conditions. Corrosion was confirmed in locations where the cover concrete was spalled or dust appeared through the cracks. The cover depth was measured using a non-destructive cover meter with electromagnetic pulse induction technology, as shown in Figure 1. In the deteriorated members, the cover depth was measured at least three times within approximately 200 mm of the spalling or cracking. Additionally, this depth was measured at three randomly selected locations on the members without any signs of deterioration.

In Japan, 24 short-span bridges (2.4–14.8 m) located in Uwajima city were visually inspected in 2017, with a focus on short-slab deterioration. In Thailand, 61 bridges were randomly inspected in Bangkok, and the cover depths of 20 bridges were measured at points where physically possible. In Vietnam, 36 bridges surrounding Hanoi were inspected by measuring the cover depth. The survey areas for each country and location climate information are shown in Figure 2 and Table 1, respectively. Uwajima is located in the temperate zone to have winter with snow, while the Bangkok and Hanoi vicinities are located in the tropical and subtropical zones with frequent squalls in rainy seasons, respectively. Some bridges in the Uwajima and Hanoi vicinities can be affected by airborne chloride ions, while bridges in Bangkok far from the seashore are not attacked by chloride ions. The effect of climates and boundary conditions on the deterioration was compared.

### 2.2. Typical Deterioration and Relationship between Corrosion-Induced Deterioration and Cover Depth

Typical examples of bridge deterioration owing to corrosion are shown in Figure 3. The deterioration in the slabs of the short-span bridges in Japan was mainly attributed to a small cover depth, which resulted in spalling. In Thailand and Vietnam, the ends of the girders were mostly deteriorated due to water seepage from the supply lines, and spalling in the slab similar to that in Japan was also observed. Deterioration related to water seepage is often found in Japan; however, no significant difference in the corrosion-induced deterioration was observed according to climate change.

The relationship between the cover depth and corrosion-induced deterioration, such as cracking or spalling, is shown in Figure 4. Regardless of the member type, a minimum cover depth of 40 mm prevented steel bar corrosion in Japan and Vietnam; however, a few cases of corrosion were observed in Thailand even when the cover depth was greater than 40 mm. Figure 5 shows the relationship between the construction year and cover depth with or without corrosion for the bridges in Uwajima city and around Hanoi, whose construction years were obtained. The date of construction of bridges in Thailand could not be obtained. The water penetration calculation in the practical design using the Lucas–Washburn theory [12] reveals that moisture can infiltrate deeper into concrete with an increase in service life, eventually reaching the reinforcing steel, leading to corrosion. According to the design, corrosion should proceed to cause spalling or cracking in older structures with deeper water penetration; however, this is not observed in real structures. It is noted in this study that spalling and cracking with the observed corrosion or rust were found in some members with cover depths of less than 40 mm, regardless of the construction year and environment in both Japan and Vietnam.

Based on the survey results, it is hypothesised that water penetration in real structures under drying and wetting conditions with rainfall would not exceed approximately 40 mm unless the cover concrete was very porous or had cracks, which were not included in our survey. Unfortunately, the survey was always carried out on sunny days because of limited time. Thus, it was impossible to clearly identify the water penetration at the measurement points during periods of rain. In addition, because the carbonation depth was also not measured owing to the difficulty in obtaining permission for coring of public bridges, a previous field survey with the detailed information is referred to in the next section to discuss the hypothesis.

### 2.3. Relationship among Cover Depth, Carbonation Depth, and Spalling in Previous Field Surveys

Similar surveys focusing on water supply and carbonation depth were reported in Japan. Ishibashi et al. [10] measured the cover depth and carbonation by drilling the cover concrete depth in 95 railway structures (mainly viaducts) with service lives ranging 15–80 years. Most of the structures were located far from the seashore. Figure 6, which is reproduced by the authors based on [10], shows the relationship between spalling, carbonation depth, and cover depth with and without water supplied by rainfall or other means. Spalling due to corrosion is mostly prevented when the cover depth is more than 30 mm regardless of the carbonation depth and age, even in the presence of water. Spalling was observed in members with a thin cover depth of less than 5 mm, even in the absence of water.

Maehara and Iyoda [11] also surveyed cracking and spalling and measured cover depth and carbonation depth by chipping the cover concrete in 36 real structures without external chloride ion supply, such as bridges, walls, tunnels, and others in Japan. The ages ranged from 11 to 87 years at the time of the survey. Figure 7 shows the relationship between carbonation depth, cover depth, and spalling, according to the apparent corrosion level [22], which was replotted by the authors focusing on water supply. In the survey, the members with a cover depth of more than 40 mm had little spalling and slight corrosion, regardless of the carbonation depth, water supply and construction year, similar to the results of the survey by Ishibashi et al. [10].

The above surveys also suggested that an adequate cover depth of over 40 mm can prevent spalling by corrosion, even though porous cover concrete with a low strength of less than 15 MPa or a high water-to-cement ratio (W/C) of over 55% was included in the surveyed structures. Carbonation in members without exposure to water can cause corrosion, although not severe enough to cause spalling. While severe corrosion causing spalling was observed mainly in members with a cover depth of less than 40 mm and water supply. Our survey indicated the importance of adequate cover depth to prevent spalling or cracking by steel corrosion even in the coastal areas of Uwajima city and near Hai-Phong with airborne chloride ions, which were not included in previous studies [10,11]. Hence, water penetration in concrete under drying and wetting conditions, which can be the most important factor causing corrosion deterioration as the specification indicates [12,13], was experimentally studied to examine the possibility of water stagnation during infiltration.

## 3. Experimental Study of Moisture Penetration in Concrete under Drying and Wetting Conditions

### 3.1. Mix Proportion and Specimens

Two types of concrete were prepared to examine the effect of W/C on water penetration. The mix proportions are shown in Table 2, assuming dense concrete with W/C = 0.45 and porous and low-strength concrete with a W/C of 0.60 as the worst case. In the case of a lower W/C (i.e., 0.45), a chemical admixture was used to obtain the desired workability, but no admixture was used for W/C = 0.60 to avoid segregation resulting from high workability.

Concrete specimens with cover depths of 10, 30, and 40 mm, with heights of 56, 76, and 86 mm, respectively, were cast. The heights were selected to maintain the same depth as the reinforcement. All specimens had lengths and depths of 150 and 76 mm, respectively, with D16 steel in the centre to maintain the previously specified cover depths. To investigate the water penetration at depths of 10, 30, and 40 mm from the surface, the electrical resistivity was measured horizontally at each depth based on the four-point (Wenner probe) method. A soft copper wire with a diameter of 1.6 mm was used as an electrode. The wires were inserted to a depth of approximately 20 mm, the ends were exposed from the side of the specimens using a heat-shrinkage rubber tube after covering all other surfaces, and the interval between the electrodes was 30 mm. Figure 8 shows a schematic representation of the specimens.

### 3.2. Experimental Program

The specimens were cured for 4 days using sealed wooden moulds. After removing the wooden formwork, the surface opposite to the casting surface to be exposed to drying and wetting was sealed with aluminium tape, whereas the other surfaces were coated with epoxy resin to prevent moisture absorption and evaporation during the experiment. At the age of 5 days, after hardening of the resin, the aluminium tape was removed, and the bottom of the specimen without sealing was immersed in a shallow water pool with a depth of a few centimetres for 1 day, simulating wetting conditions soon after the curing. From the age of 6 days, drying was started. The specimens were dried at 40 °C under 60 ± 2 % RH in the chamber, simulating the most severe drying in hot regions such as Thailand. The specimens with W/C = 0.6 were also dried at 20 °C under 60 ± 5% RH in a climate-controlled room, which is often specified in the Japan Industrial Standard as the standard drying condition for concrete [23]. After drying for 6 days, water was supplied from the bottom of the specimen as explained previously for 1 day at 20 °C because water-induced spalling is often found at the bottom of the member. To examine water penetration after drying, the current and voltage before and after wetting were measured to calculate the specific resistance based on the four-point method. After two cycles of drying and wetting, the specimens were dried for 13 days and then exposed to water for 3 more days to examine the effect of a longer duration of drying and wetting. The drying and wetting processes are shown in Figure 9. The weight was also measured, before and after wetting, to examine the mass loss and gain due to drying and wetting.

The electrical specific resistivity was obtained to examine the water content at each depth by measuring the voltage and current in the electric wire in the specimen before and after wetting, based on the four-point (Wenner probe) method. The experiment to focus on the change in resistance due to water penetration can be utilised to discuss the one-dimensional water penetration process at each depth. An electric diagram example is shown in Figure 10. A voltage of 5 V and a sine wave of 100 Hz were used as the alternators. The specific resistance *ρ* at each depth was calculated as follows:(1)$$\rho =2\pi a\frac{V}{I}$$
where *a*, *V*, and *I* are the distance between the electrodes, voltage, and current in the line, respectively. The current in the line was obtained by measuring the voltage at a shunt resistance of 330 Ω, whereas the voltage was measured between two poles at the centre. The voltages were recorded after the values were stable. The average specific resistance values of the two specimens at each depth were obtained.

### 3.3. Results

The change ratios of specific resistance before and after wetting for 1 day in the 1st and 2nd cycles are shown in Figure 11, as the discussion based on the change ratios can be clearer than that based on the variation in the specific resistance itself. The absolute values of specific resistance variation are presented in Appendix A. Each specimen was identified with respect to the drying conditions (H = 40 °C, N = 20 °C), water-to-cement ratio (W/C = 45%: W45, W/C = 60%: W60), and cover depth (10 mm: C10, 30 mm: C30, 40 mm: C40). For instance, concrete with W/C = 45% and cover depth of 30 mm dried under an RH of 60% at 40 °C is termed “HW45C30”. The ratio of specific resistance was obtained by dividing the difference in the specific resistance before and after wetting by the specific resistance before wetting. When the water penetrates around the position of the resistance measurement, the resistance can be reduced because of the lower resistance of water, resulting in a negative change ratio before and after wetting.

In the case of low W/C, the specific resistances at various depths, except those of HW45C10 and HW45C30 at a depth of 10 mm in the 2nd cycle, increased even after wetting from the surface for 1 day. This could be because the progressive hydration due to the low W/C makes the pore structure denser with ageing, especially at earlier ages. This substantially prevented water penetration and consumed water in the saturated pores for hydration, even during wetting, which increased the resistance. In contrast, the specific resistances of the concrete with a W/C of 0.6 were reduced in proportion to an increasing depth up to 30 mm because of the 1 day wetting after drying at 40 °C. In addition, the specific resistances of the specimens dried at 20 °C were reduced at each particular depth after 1 day of wetting which suggested the possibility of water penetration at steel. However, the resistance at depths of 30 and 40 mm could be reduced even by water infiltration around the dried surface at 10 mm under the concentric current flow around the measurement positions if the water at depths greater than 30 mm was not severely dried by mild drying at 20 °C and 1 day of wetting as discussed later.

Figure 12 shows the change ratio of the specific resistance before and after wetting for 1 and 3 days in the 3rd cycle. The change in specific resistance due to wetting for 1 day, even after a longer drying period, was similar to that of the 1st and 2nd cycles after drying for 6 days, as shown in Figure 11. The resistance decreased with an increase in the wetting time in all specimens. However, the changes at 30 and 40 mm depths for specimens HW45C30 and HW45C40 with low W/C were still positive even after wetting for 3 days, indicating that no water infiltration occurred until a depth of 30 mm due to the dense pore structures and the effects of self-desiccation. In the specimens with W/C = 60%, the resistance reduction ratios from 1 day to 3 days in the wetting process were larger at depths of 30 and 40 mm when dried at 40 °C, than when dried at 20 °C. This suggested that the water gradually penetrated deeper through unsaturated pores with an increase in the wetting period after drying at higher temperatures, whereas the water could not penetrate in less unsaturated pores, indicating a slight effect of a long wetting period on the resistance reduction when dried for 13 days at 20 °C.

### 3.4. Discussion

According to the change ratios of specific resistance at various depths of specimens before and after wetting for 1 and 3 days, it is suggested that the water does not reach the steel position at a depth of 40 mm in concrete with W/C = 45% after drying at 40 °C. In the case of higher W/C = 60%, the water may penetrate at a depth of 40 mm through unsaturated pores with an increase in the wetting period, especially after severe drying at 40 °C, but it is plausible that the water penetration at a deeper depth over 30 mm might be inhibited through less unsaturated pores after less severe drying at RH = 60% at 20 °C.

To explicitly examine the water penetration at each depth, the specimens after the 3rd cycle were dried again under the same drying conditions for 6 days, and the mass loss due to oven drying at 105 °C was measured at each depth before and after wetting for 3 days. One side of the specimens with a cover depth of 40 mm was sliced with a cut approximately 20 mm thick and perpendicular to the drying and wetting surface using a concrete cutter. Subsequently, the sliced samples were cut at 20 mm intervals from the top and dried at 105 °C to measure the moisture content at each depth.

Figure 13 shows the moisture content measurement results. The moisture of the specimen with a W/C of 45% did not increase at depths of 20–40 and 40–60 mm after wetting for 3 days; whereas, an increase in the moisture content of specimens with 0–20 mm depth was observed due to water penetration, as indicated by the electric resistance. This was attributed to the dense pore structure, which requires a longer time for water to penetrate deeply. In contrast, in specimens HW60C40, the moisture content at any depth increased markedly, indicating that water penetrated to an approximate depth of 40 mm from the surface by wetting for 3 days after drying at 40 °C. Concrete with W/C = 60% has coarser pore structures, which could allow water migration largely owing to the capillary suction when dried, resulting in an increased moisture content up to 40–60 mm after 3 days of wetting. However, the water did not reach depths of approximately 40 mm in only 1 day of wetting after drying at 40 °C for 6 days because the electric resistance did not decrease before and after 1-day wetting, except in the case of HW60C10, as shown in Figure 11. When the concrete was dried at RH = 60% at 20 °C for 6 days, the moisture content before wetting at depths of 20–40 and 40–60 mm did not differ, indicating that the pores at depths greater than 20 mm would not be dried under mild drying conditions for 6 days. Even when water was introduced from the surface after drying, the moisture content at depths greater than 20 mm did not change because the water could not infiltrate deeper through the saturated pores, whereas the electric resistance was reduced after wetting, as shown in Figure 11.

As assumed previously, the resistance of the NW60 specimens at depths of 20 and 40 mm would be influenced by water penetration into the surface to approximately 20 mm, even though the moisture content did not change at further depths. This implied that mild drying at 20 °C might take a longer time to remove water from the pores deeper than 20 mm and permits water infiltration only near the surface with less unsaturated pores.

As shown in Figure 11 and Figure 12, the smaller cover depth exhibited a higher resistance reduction at any depth after wetting, which indicates more and deeper water penetration. Figure 14 shows the change in mass after drying for 6 d. The specimens with smaller cover depths lost more water during drying. Although the specimens of HW60C10 dried at 40 °C had a slight change in mass compared to HW60C30 and HW60C40, which have larger cover depths, the result may contain some errors because the mass change due to drying was smaller than that of HW45C10 with the same dimensions, which have denser pore structures. Eddy et al. [24] reported that the pore structure becomes coarse with a thinner concrete cover. These results can be attributed to a coarser pore structure, which has a smaller resistance to water infiltration and more moisture loss during drying. Hence, concrete with a smaller cover depth allows more water penetration owing to the coarser pore structures.

The experiment to study the water penetration under wetting and drying conditions revealed that the water penetration at a depth of 40 mm could be inhibited, especially in the concrete with lower W/C and larger cover depth and the penetration depth by wetting is dependent on the pore saturation after drying. Hence, the numerical analysis was carried out to investigate the internal moisture states in pores according to drying conditions and period.

## 4. Discussion of Experimental Results Using Numerical Analysis of Drying Process

To quantitatively investigate the internal moisture states in pores during the drying process in the above experiment, the variations in pore humidity and the saturation of pores, such as capillary and gel pores, due to drying under RH = 60% at 20 and 40 °C were numerically calculated. The thermodynamic simulator for concrete, named DuCOM (Durability of Concrete Model, Ver. 5.11 2007 model, Tokyo), developed by the University of Tokyo was used. DuCOM consists of multicomponent hydration, a micropore development structure, and a moisture equilibrium/transport model [25,26].

The calculated pore distribution of the tested concrete after 5 days of sealed curing is shown in Figure 15. Concrete with W/C = 45% has a denser pore structure than concrete with W/C = 60%, especially for pore radii above 100 nm, which comprises mainly capillary pores. This is because more hydrates precipitated in concrete capillary pores with lower W/C, according to the definition of capillary pores in DuCOM [27].

DuCOM can determine the internal moisture states of concrete with any mix proportion and size under various drying conditions, including high temperatures [28]. The simulation focused on the moisture states from the dried surface along with the depth for different drying times and temperatures because a detailed discussion of water penetration into unsaturated pores by drying requires a consistent experiment. Additionally, model verification is not possible because of limited experimental results. Hence, the drying process after sealed curing for 5 days and wetting for 1 day with a simple assumption of a boundary RH = 99.5% at the wetting surface was analysed from the start of drying to 30 days of drying without wetting using water.

The variations in internal pore humidity, total water content with physically evaporable water and chemically bound water, and degree of saturation in the capillary and gel pores defined by DuCOM due to drying under RH = 60% at 20 and 40 °C are shown in Figure 16. The pore humidity and water content were gradually reduced from the drying surface with increasing drying time; however, they decreased insignificantly at depths greater than 40 mm even after 13 days of drying. The humidity and water content reduction by drying was more significant at higher temperatures for both types of concrete. The gradual reduction in the pore humidity in the concrete with W/C = 45% is also attributed to the consumption of water for hydration (self-desiccation) even at depths greater than 40 mm. The total water content with chemically bound water was almost the same at depths of over 20 mm after drying for 6 days at 20 °C, while the water content was reduced at depths of 20–30 mm after drying for 6 days at 40 °C, especially in the case of W/C = 60%. The calculated results are comparable to the measured moisture contents, as shown in Figure 13, which indicates no difference at depths of 20–40 and 40–60 mm after drying for 6 days at 20 °C and an increase in moisture content along with the depth after drying at 40 °C. The degree of saturation in the capillary pores also gradually decreased from the surface with drying, whereas that in the gel pores was reduced only around the drying surface, and it was maximum at a depth of 15 mm in the case of W/C = 45%. The capillary pores can be slightly desiccated along with a depth of more than 30 mm when dried for 13 days at 40 °C, whereas the degree of saturation remains almost the same along with the depth for over 20 mm in the case of concrete with W/C = 60% dried at 20 °C. Although the capillary pores at depths beyond 30 mm are partially dried when exposed to 40 °C for 13 days, as indicated by the simulation, it is assumed that it would be difficult for water from the wetting surface to penetrate to a depth of more than 30 mm in the lower W/C concrete with fewer capillary pores, as shown in Figure 15.

When drying was carried out for 30 days at 40 °C in the simulation, the pore humidity, total water content, and degree of saturation in the capillary pores at depths exceeding 40 mm can be decreased; however, at a depth exceeding 50 mm, the reduction is attributed to self-desiccation. After drying at 20 °C for 30 days, the moisture evaporated at a depth of less than 30 mm. Based on the numerical analysis for the drying of concrete with W/C = 0.45 and 0.60, it was concluded that maximum moisture evaporation in conventional concrete occurred at a depth of approximately 30–40 mm, especially in relatively large pores such as capillary pores, unless the concrete was exposed to continuous drying for more than one month. The oxygen, which is also necessary for corrosion, can easily penetrate into cover concrete through the unsaturated pores but may be difficult to diffuse at deeper depth over 40 mm where would keep high pore humidity over 90% according to the simulation.

Laboratory experiments indicated that it is difficult for water to infiltrate through dense pore structures or few unsaturated pores to a depth exceeding 40 mm. Drying of concrete is inhibited by natural periodic rainfall, as reported in a previous study [29]. The numerical simulation indicated that a significantly longer time was required to dry relatively large capillary pores at depths of more than 40 mm, even when dried for one month under RH = 60% at 40 °C. Although the pores several centimetres deep became partially unsaturated under severe drying, such as at an elevated temperature of 40 °C, the wetting duration of a few days of rainfall may not be sufficient for water to penetrate the reinforcement with a cover depth of approximately 40 mm, as suggested in the described experiment. It is also indicated that the surface region of concrete exposed to water is not instantaneously saturated, and the water uptake slows with an increase in moisture content [30]. It would support the survey findings that the members with a cover depth of more than 40 mm were unlikely to have the corrosion-induced spalling in Thailand with high ambient temperature all seasons and frequent squall. Additionally, the experimental and numerical studies suggested that the mild drying condition with RH = 60% at 20 °C corresponding to the approximate average RH and temperature in Japan and Vietnam cannot cause severe drying in pores of the cover concrete at depths of greater than 40 mm when the water is frequently supplied. A high degree of saturation in the cover concrete can prevent oxygen penetration through unsaturated pores, which inhibits corrosion.

In conclusion, it is difficult for both water and oxygen to not infiltrate the reinforcing steel of concrete with adequate cover depth, such as 40 mm, and to cause severe corrosion, as indicated in the field survey in Japan, Vietnam and Thailand with different climates. Li et al. [31] numerically investigated the influential depth of moisture transport in concrete under drying-wetting conditions and suggested an adequate depth of approximately 30 mm, which is reasonably consistent with our study. A more consistent and quantitative discussion based on both numerical and experimental approaches is necessary to determine the effective cover depth to inhibit corrosion caused by water penetration as well as oxygen diffusion. The preliminary results of this study emphasised the importance of an adequate concrete cover depth for the prevention of deterioration due to corrosion based on both on-site and laboratory investigations.

## 5. Conclusion

In this study, the corrosion-induced deterioration of reinforced concrete bridges in Japan, Thailand, and Vietnam was visually surveyed, and the cover depth in the members around the deterioration was measured and compared to those without deterioration. In addition, the water infiltration in specimens with different cover depths under drying and wetting cycles was examined to measure the specific electric resistance. The findings and suggestions of this study are summarised as follows:

(1) Corrosion-induced deterioration in bridges was frequently observed where water was continuously supplied from rainfall and other means, regardless of climatic region and member type.

(2) Reinforced concrete members with cover depths exceeding 40 mm, except in a few cases, were protected from visible spalling by corrosion in all surveyed countries, regardless of the boundary conditions and construction year.

(3) It was experimentally found that water penetration, owing to wetting for a few days after drying for approximately 1 or 2 weeks, cannot exceed 40 mm in the case of W/C = 45% and mild drying at 20 °C.

(4) The numerical simulation indicated that significant time (>1 month) was required to dry relatively large capillary pores at depths of more than 40 mm. It is suggested that the capillary pores in the cover concrete with a depth of over 40 mm may be mostly saturated owing to the natural periodic rainfall, leading to the difficulty of water and oxygen penetration into the reinforcement to cause corrosion, despite an increase in service life as indicated in the field survey.

## Figures and Tables

**Figure 1 materials-14-03478-f001:**
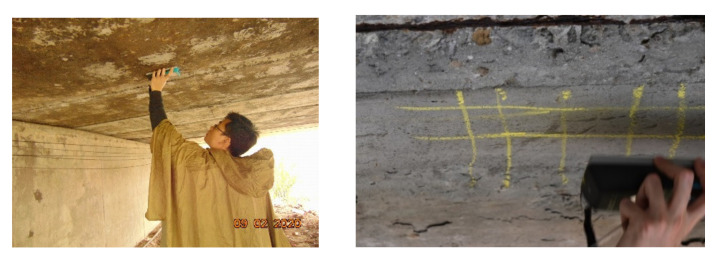
Cover depth measurement.

**Figure 2 materials-14-03478-f002:**
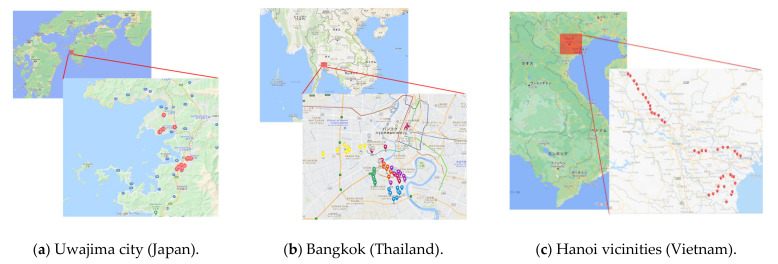
Survey locations in the different countries (Map data from Google 2020) (**a**) Uwajima city (Japan) (**b**) Bangkok (Thailand) (**c**) Hanoi vicinities (Vietnam).

**Figure 3 materials-14-03478-f003:**
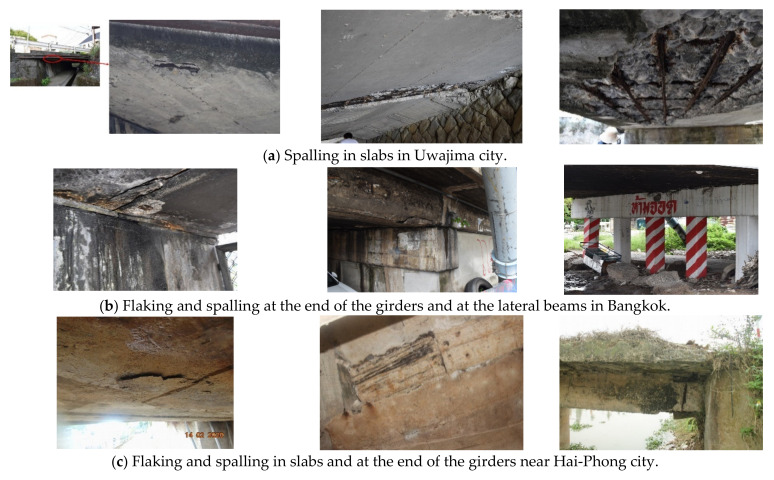
Typical examples of corrosion-induced deterioration in bridges in each selected country. (**a**) Spalling in slabs in Uwajima city (**b**) Flaking and spalling at the end of the girders and at the lateral beams in Bangkok (**c**) Flaking and spalling in slabs and at the end of the girders near Hai-Phong city.

**Figure 4 materials-14-03478-f004:**
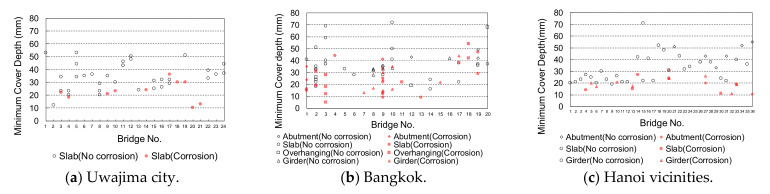
The relationship between cover depth and corrosion for bridges in each selected country. (**a**) Uwajima city (**b**) Bangkok (**c**) Hanoi vicinities.

**Figure 5 materials-14-03478-f005:**
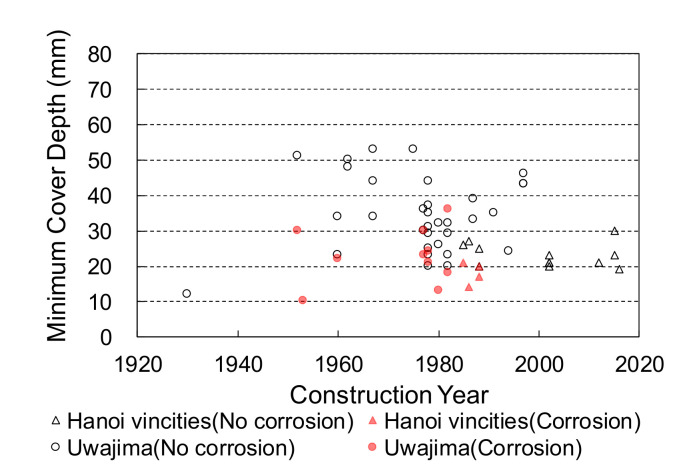
The relationship between construction year and minimum cover depth, with and without corrosion, in Uwajima city (Japan) and the Hanoi vicinity (Vietnam).

**Figure 6 materials-14-03478-f006:**
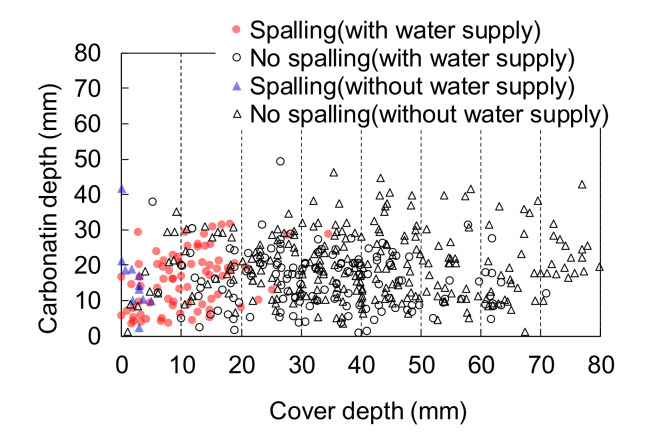
The relationship between carbonation depth, cover depth, and spalling, due to the corrosion with and without water supply [10] (replotted by authors).

**Figure 7 materials-14-03478-f007:**
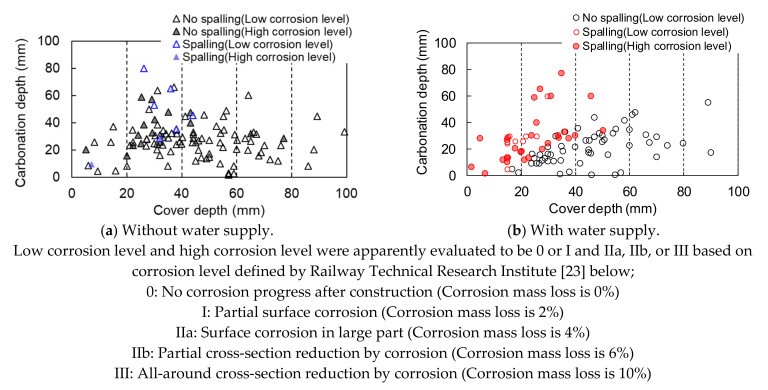
The relationship among carbonation depth, cover depth, and spalling, according to corrosion level with and without water supply [11] (replotted by authors). (**a**) Without water supply (**b**) With water supply.

**Figure 8 materials-14-03478-f008:**
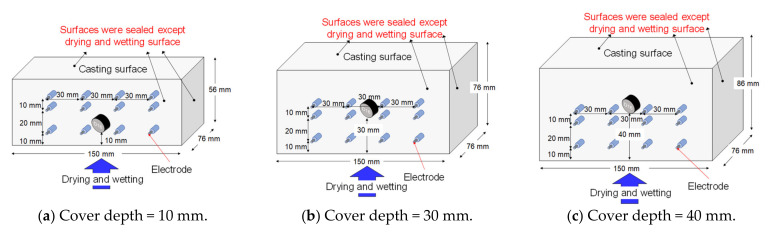
A schematic representation of specimens with inserted electrodes. (**a**) Cover depth = 10 mm (**b**) Cover depth = 30 mm (**c**) Cover depth = 40 mm.

**Figure 9 materials-14-03478-f009:**
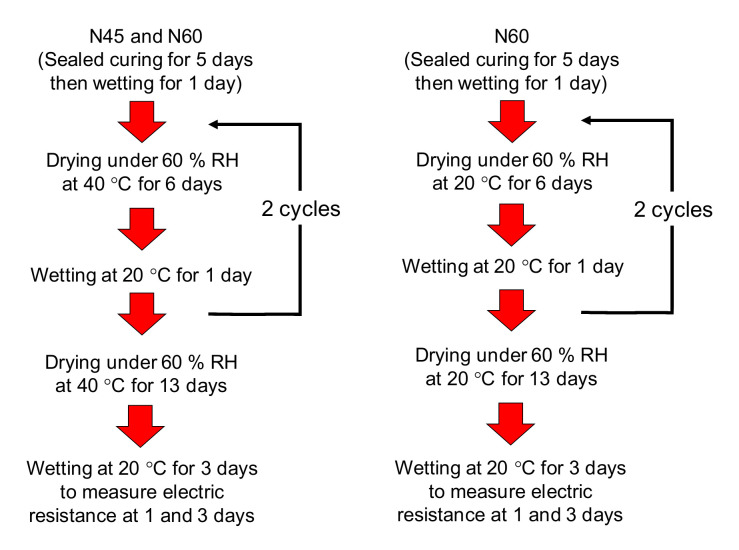
Summary of drying and wetting processes.

**Figure 10 materials-14-03478-f010:**
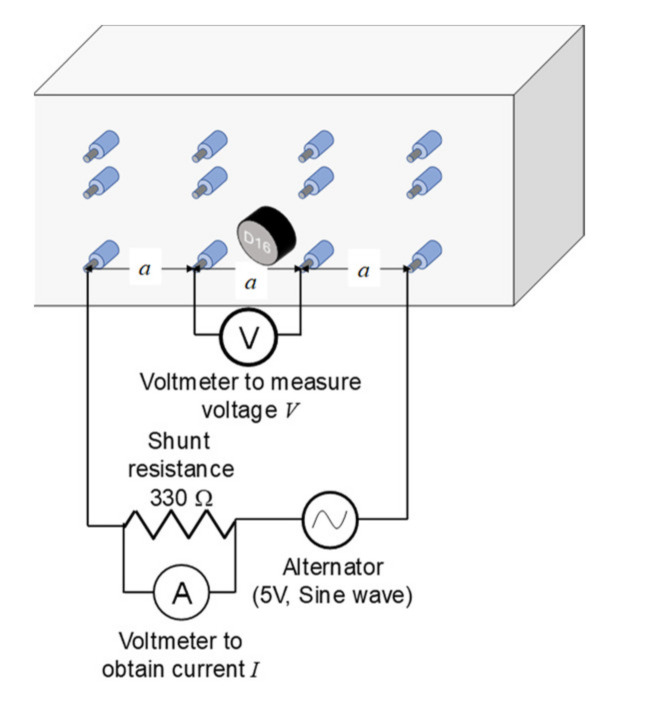
A schematic of the electric circuit for the specimen with a cover depth of 10 mm.

**Figure 11 materials-14-03478-f011:**
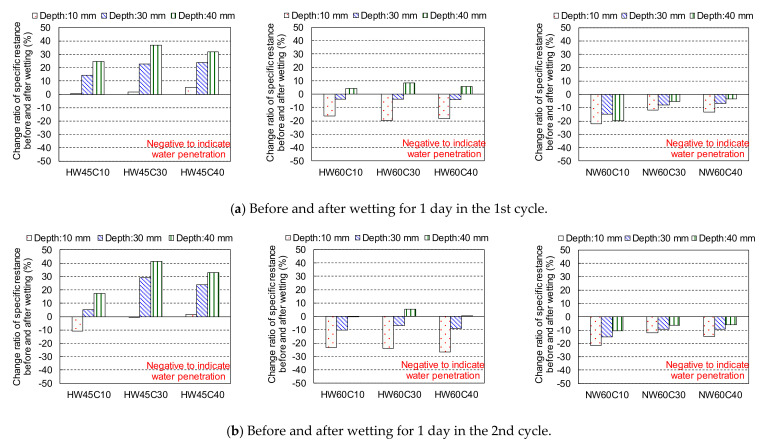
The change ratio of specific resistance before and after wetting for 1 day after sealed curing in the 1st and 2nd cycles. (**a**) Before and after wetting for 1 day in the 1st cycle (**b**) Before and after wetting for 1 day in the 2nd cycle.

**Figure 12 materials-14-03478-f012:**
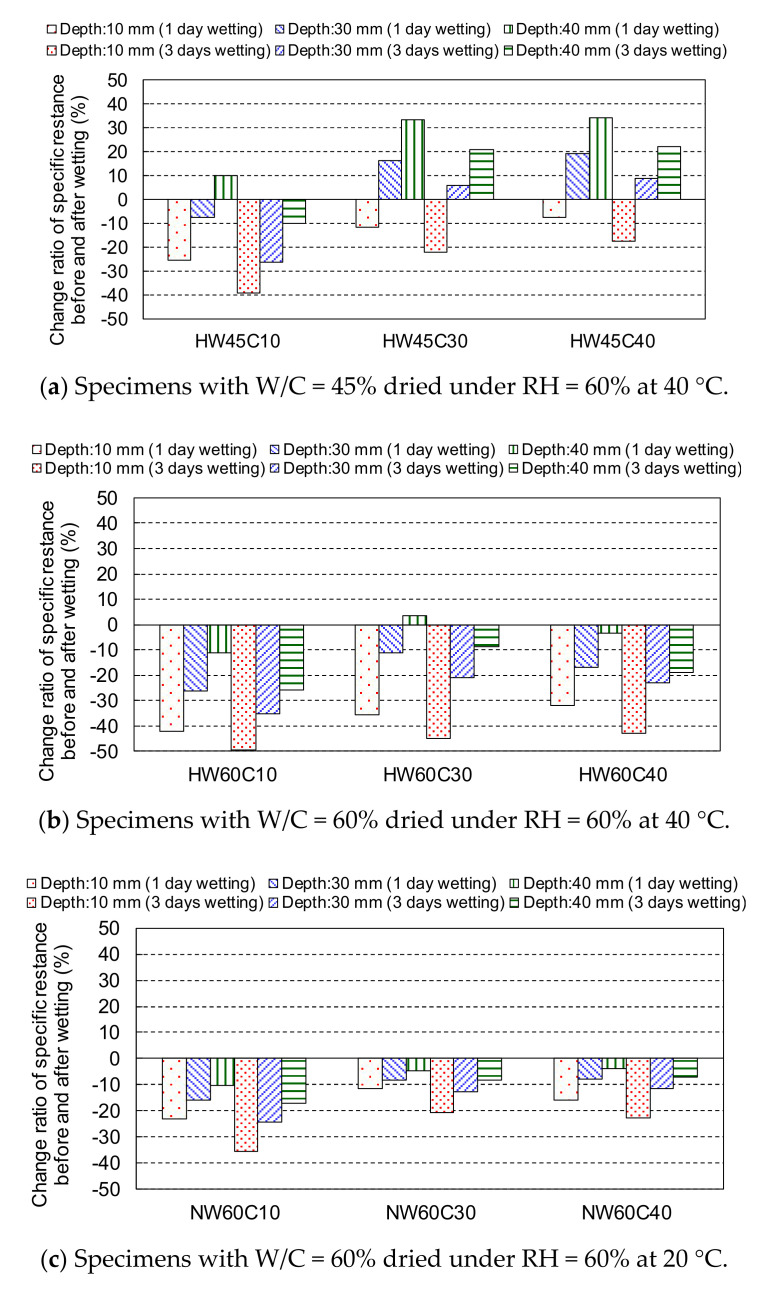
The change ratio of specific resistance before and after wetting for 1 and 3 days on the 3rd cycle. (**a**) Specimens with W/C = 45% dried under RH = 60% at 40 °C (**b**) Specimens with W/C = 60% dried under RH = 60% at 40 °C (**c**) Specimens with W/C = 60% dried under RH = 60% at 20 °C.

**Figure 13 materials-14-03478-f013:**
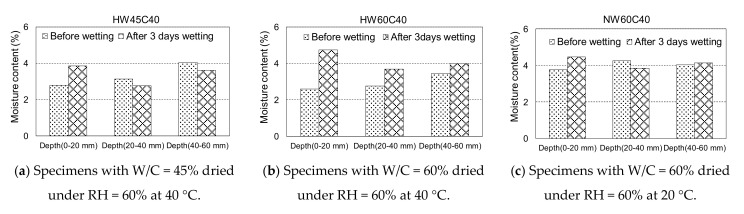
The moisture content at each depth before and after 3 days wetting for the 3rd cycle. (**a**) Specimens with W/C = 45% dried under RH = 60% at 40 °C (**b**) Specimens with W/C = 60% dried under RH = 60% at 40 °C (**c**) Specimens with W/C = 60% dried under RH = 60% at 20 °C.

**Figure 14 materials-14-03478-f014:**
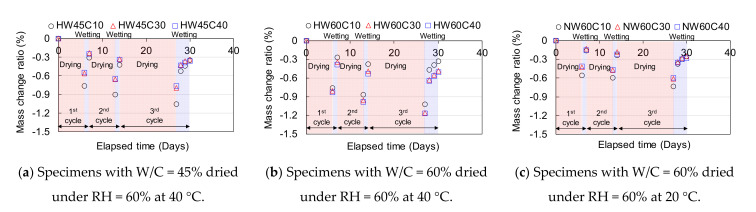
Mass loss after drying at the age of 6 days. (**a**) Specimens with W/C = 45% dried under RH = 60% at 40 °C (**b**) Specimens with W/C = 60% dried under RH = 60% at 40 °C (**c**) Specimens with W/C = 60% dried under RH = 60% at 20 °C.

**Figure 15 materials-14-03478-f015:**
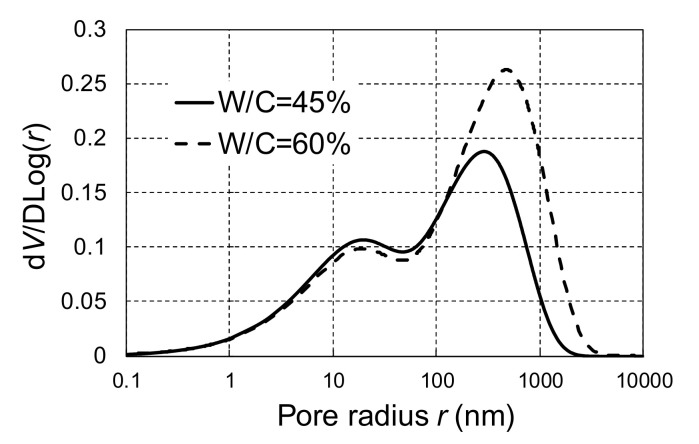
Pore distribution of concrete after sealed curing.

**Figure 16 materials-14-03478-f016:**
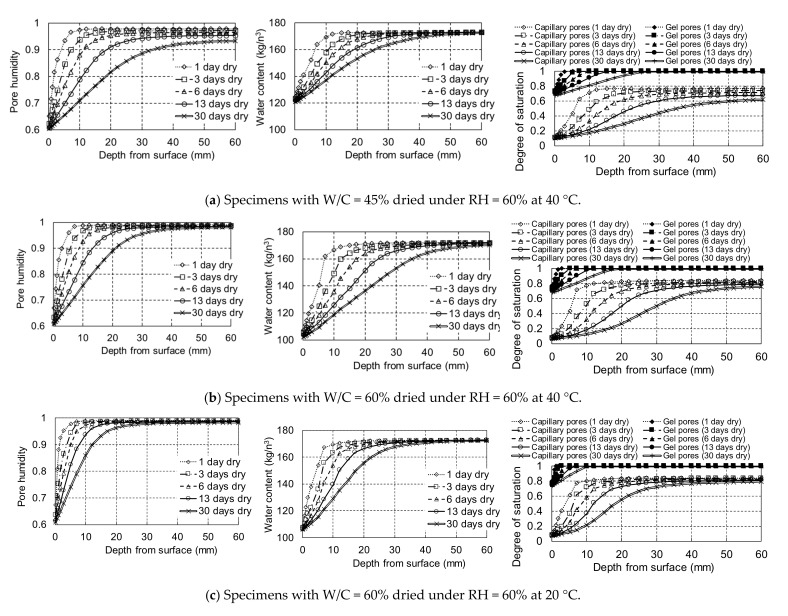
Calculated variation in pore humidity, water content, and degree of saturation with drying in each specimen using DuCOM. (**a**) Specimens with W/C = 45% dried under RH = 60% at 40 °C (**b**) Specimens with W/C = 60% dried under RH = 60% at 40 °C (**c**) Specimens with W/C = 60% dried under RH = 60% at 20 °C.

**Table 1 materials-14-03478-t001:** Climate information in each location.

	Temperature (°C)			Average Annual Humidity (%)	Average Annual Precipitation (mm)	Average Annual Number of Days with Precipitation
	Monthly Average max	Monthly Average min	Annual Average			
Uwajima	32.5	2.5	17.2	73.8	1805	142
Bangkok	34	21	28	73.8	1450	137
Hanoi	32.2	15	23	85	1607	187

The information in Uwajima was obtained by averaging data for five years (2012–2017) from the database of the Japan Meteorological Agency [20], while the information in Bangkok and Hanoi was obtained from the database of Canty and Associates LLC [21].

**Table 2 materials-14-03478-t002:** Mix proportion.

Mix ID	W/C (%)	Water (kg /m^3^)	Cement (kg/m^3^)	Fine Aggregate (kg/m^3^)	Coarse Aggregate (kg/m^3^)	Air-Entraining and Water-Reducing Agent (mL/m^3^)	Air-Entraining Agent (g/m^3^)
N45	45	172	383	755	1003	958	23
N60	60	172	287	843	993	-	-

Cement: Ordinary Portland cement (density = 3.16 g/cm^3^), Fine aggregate: River sand (saturated surface dry density = 2.59 g/cm^3^), Coarse aggregate: Crushed gravel (maximum size = 20 mm, saturated surface-dry density = 2.71 g/cm^3^).

## Data Availability

The data is contained within the article.

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
