# Peer review of "The Cover Depth Effect on Corrosion-Induced Deterioration of Reinforced Concrete Focusing on Water Penetration: Field Survey and Laboratory Study"

_materials, 2021, doi:10.3390/ma14133478_

Round 1

Reviewer 1 Report

Notes on the article of Shingo Asamoto, Junya Sato, Shinichiro Okazaki, Pang-jo Chun, Raktipong Sahamitmongkol, Giang Hoang Nguyen “The cover depth effect on corrosion-induced deterioration of reinforced concrete focusing on water penetration: Field survey and laboratory study”

The paper reports results of studying the ability of reinforced concrete bridges to resist spalling or cracking arising from corrosion. In addition, the impact of climate on corrosion induced deterioration in bridges was studied. The authors established then the cover depth of 40 mm is effective for preventing of water penetration under natural conditions and corrosion at the position of the steel.  The results of this article have the high practical importance. This is an interesting and well-written report, which should be published after minor revisions that are listed below:

1) The authors did a lot of practical and experimental work and correctly described their research. However, the article itself looks more like an experimental report. Authors should more carefully work out the discussion part of the article.

2) In their conclusions, the authors state: «Members with cover depths exceeding 40 mm, except in a few cases, were protected from spalling by corrosion in all surveyed countries, regardless of the boundary conditions and construction year». It is not entirely clear if the authors mean spalling by corrosion of concrete or steel structures underneath. If spalling by corrosion of steel structures was meant, then it is not clear how the degree of spalling was assessed.

Author Response

Response to Reviewer 1 Comments

Thank you for your kind reviewing our manuscript and giving pertinent comments. They helped us to substantially improve the manuscript in many aspects.

We tried to answer all the comments raised in your report as follows.

Point 1: The authors did a lot of practical and experimental work and correctly described their research. However, the article itself looks more like an experimental report. Authors should more carefully work out the discussion part of the article. 
Response 1: Thank you for your kind comments. As other reviewers also pointed out similar issues, we added some discussion in chapters 3 and 4.
Modification: Chapters 3 (Section 3.4) and 4.

Point 2: In their conclusions, the authors state: «Members with cover depths exceeding 40 mm, except in a few cases, were protected from spalling by corrosion in all surveyed countries, regardless of the boundary conditions and construction year». It is not entirely clear if the authors mean spalling by corrosion of concrete or steel structures underneath. If spalling by corrosion of steel structures was meant, then it is not clear how the degree of spalling was assessed.
Response 2: Thank you for your valuable comment. As you pointed out, the explanation was unclear to cause readers’ misunderstanding. In our survey, the spalling due to the corrosion only in reinforced concrete (no steel structures) was visually identified based on the finding of steel rust and corrosion around the spalling and the degree was not assessed clearly. The explanation was modified according to your comments.
Modification: Conclusion

Reviewer 2 Report

This article aims to survey field and laboratory study about the cover depth effect on corrosion-induced deterioration of RC bridges focusing on water penetration. There are following questions:

  1. Abstract need to be rewritten to report about the main and new findings obtained in this paper briefly, not just general known knowledge.
  2. The topic of this research is generally known about the mechanism and knowledge of steel corrosion, but the author's contribution and novelty are not enough emphasized.
  3. Why choose the bridge between these three countries, and what is the special significance.
  4. The author should supplement the correlation between the laboratory simulation conditions and the survey field.
  5. The authors should emphasize the similarities and differences in the cited articles, and summarize their own views and contributions.
  6. The author should discuss the results of the study in more depth.
  7. The conclusions do not add to current knowledge. Authors have to better to condense the conclusions.

Author Response

Response to Reviewer 2 Comments

Thank you for your kind reviewing our manuscript and giving pertinent comments. They helped us to substantially improve the manuscript in many aspects.
We tried to answer all the comments raised in your report as follows.

Point 1: Abstract need to be rewritten to report about the main and new findings obtained in this paper briefly, not just general known knowledge.
Response 1: Thank you for your instruction. As you pointed out, we noticed the abstract conclusion was just general in concrete engineering. We modified the abstract to focus on our practical, experimental and numerical findings.
Modification: Abstract

Point 2: The topic of this research is generally known about the mechanism and knowledge of steel corrosion, but the author's contribution and novelty are not enough emphasized.
Response 2: As you pointed out, the topic of cover depth on corrosion is not new and novel but other previous studies have mainly focused on the carbonation and chloride ingress in cover concrete. In our case, the importance of water penetration on the corrosion was focused on according to the survey in different countries. We tried to explain our significance at the end of “Introduction”. Thank you for your valuable comments.
Modification: Introduction

Point 3: Why choose the bridge between these three countries, and what is the special significance.
Response 3: As explained in “Introduction”, the water penetration rate can be affected by climate. It was expected that the deterioration of reinforced concrete by corrosion would be accelerated in tropical regions due to rapid evaporation at high temperature and periodic water penetration by frequent rainfall. Even though the previous survey in Japan found the threshold of cover depth to cause corrosion-induced spalling, it was not clear whether the similar findings could be obtained in tot regions. The climate difference was explained and focus in the survey was added in the revised manuscript.
Modification: Introduction, at the end of Section 2.1

Point 4: The author should supplement the correlation between the laboratory simulation conditions and the survey field.
Response 4: Thank you for your critical comments and we carefully read the manuscript again to totally agree with your comments. We added some explanation to explain the correlation between the laboratory and numerical studies and field survey in Chapters 3 and 4.
Modification: Chapters 3 and chapter 4.

Point 5: The authors should emphasize the similarities and differences in the cited articles, and summarize their own views and contributions.
Response 5: We added some explanation about the similarities and differences from the cited papers and classical knowledge, especially in “Introduction” and section 2.3.
Modification: Introduction and section 2.3

Point 6: The author should discuss the results of the study in more depth.
Response 6: Thank you for your invaluable comments. As other reviewers also pointed out similar issues, we added some discussion in chapters 3 and 4.
Modification: Chapters 3 (Section 3.4) and 4.

Point 7: The conclusions do not add to current knowledge. Authors have to better to condense the conclusions.
Response 7: Thank you for your instruction. We also noticed some general knowledge in the Conclusion. Hence, we modified the conclusion to exclude the general knowledge and emphasize our findings.
Modification: Conclusion

Reviewer 3 Report

The present work is devoted to the study of the bridges deterioration caused by internal steel corrosion. The bridges were visually surveyed in three Asian countries to study the deterioration under different climates, focusing on the concrete cover depth. In addition, the authors performed laboratory tests to study water penetration in concrete under drying and wetting conditions.

I give a great importance to studies on the real constructions. I would like to appreciate this paper since practical results. The work is interesting and worth of publication. In my opinion the manuscript is well written, sufficiently detailed, the methods and results are adequately described and discussed. However some changes are required. Below my specific comments, in the order of their occurrence in the manuscript.

1) Keywords
In my opinion, the most important keywords related to the research objects are missing, i.e. reinforced concrete, bridges, reinforcement corrosion

2) line 43
The subscript is missing in the O2 notation.

3) lines 479-480
There is a mistake in this sentence: there is no electron flow in the concrete. Concrete is an electrolyte, so the electric charges flowing in the concrete are ions, not electrons. 
The electrons flow in the metals and do not flow into the electrolyte. The current between the metal electrodes and the concrete flows in such a way that chemical reactions of oxidation take place on the surface of one electrode, and reactions of reduction take place on the other electrode (electrons are involved in these reactions), and ions flow in the electrolyte. 

Author Response

Response to Reviewer 3 Comments

Thank you for your kind reviewing our manuscript and giving pertinent comments. They helped us to substantially improve the manuscript in many aspects.
We tried to answer all the comments raised in your report as follows.

Point 1: Keywords
In my opinion, the most important keywords related to the research objects are missing, i.e. reinforced concrete, bridges, reinforcement corrosion
Response 1: Thank you for your instruction. According to you comment, we added “Reinforced concrete bridge”, “Reinforcement corrosion” in keyword.
Modification: Keyword

Point 2: line 43
The subscript is missing in the O2 notation.
Response 2: We are sorry for missing it and thank you for your finding. We modified the corresponding subscript.
Modification: Corresponding O2 at page of revised paper

Point 3: lines 479-480
There is a mistake in this sentence: there is no electron flow in the concrete. Concrete is an electrolyte, so the electric charges flowing in the concrete are ions, not electrons.
The electrons flow in the metals and do not flow into the electrolyte. The current between the metal electrodes and the concrete flows in such a way that chemical reactions of oxidation take place on the surface of one electrode, and reactions of reduction take place on the other electrode (electrons are involved in these reactions), and ions flow in the electrolyte.
Response 3: Thank you for your pointing out our mistake. I modified from “electrons can flow” to “electric current can flow” in Appendix. In other parts, we could not find any expression of “electron flow” in the manuscript but “electric current”. If it is still wrong, would you please point out again? Thank you again.
Modification: Appendix

Reviewer 4 Report

The paper deals with the influence of water penetration in reinforced concrete structures. The article consists of three parts: in the first part the authors analyse a series of bridges built in three different regions, in the second part the authors illustrate experimental tests, in the third part the authors illustrate a numerical application.

The paper deals with a topic that is extremely interesting, current and internal to the topics of the journal. The authors describe each of the parts clearly and with a lot of data.

However, the three parts, although presenting the same topics, are quite disconnected. More than a research paper, the paper seems to be the report of 3 research on the same topic. The authors need to improve the link between different parts.

In the first part, the authors compare measurements taken on bridges from different regions. Has the quality of the materials been considered?

In the third part, the authors use software to numerically simulate the amount of water penetrating the concrete. Are these numerical results comparable with the experimental results obtained in the second part?

As specified, the paper is extremely interesting but the link between the parts needs to be improved.

Author Response

Response to Reviewer 4 Comments

Thank you for your kind reviewing our manuscript and giving pertinent comments. They helped us to substantially improve the manuscript in many aspects.
We tried to answer all the comments raised in your report as follows.

Point 1: The paper deals with a topic that is extremely interesting, current and internal to the topics of the journal. The authors describe each of the parts clearly and with a lot of data.
However, the three parts, although presenting the same topics, are quite disconnected. More than a research paper, the paper seems to be the report of 3 research on the same topic. The authors need to improve the link between different parts.
Response 1: Thank you for your invaluable comments. The link between the experimental and numerical studies and the field survey was weak as other reviewers pointed out. We tried to explain the link of each topic adding some discussion and explanation in chapters 3 and 4.
Modification: Chapters 3 (Section 3.4) and 4

Point 2: In the first part, the authors compare measurements taken on bridges from different regions. Has the quality of the materials been considered?
Response 2: Actually, we did not take into account the material quality because it is difficult to take the core sample and the core quality would be scattered depending on the construction. However, regardless of the possibly scattered material quality in each country, the effect of cover depth on the corrosion-induced deterioration was similar, which means that the concrete cover with more than 40 mm can be likely to inhibit the corrosion. It may be better to add the explanation in the manuscript but we have not clearly studied the material quality in each country even though worse case with porous concrete of W/C = 60% was taken into account in the experiment. Hence, we did not emphasize the effect of different materials in the revised manuscript but slightly added relevant explanation in section 3.1 and chapter 4.
Modification: Section 3.1 and chapter 4

Point 3: In the third part, the authors use software to numerically simulate the amount of water penetrating the concrete. Are these numerical results comparable with the experimental results obtained in the second part?
Response 3: As you and other reviewers pointed out, the relation between experiment and simulation was not clearly explained. Hence, some explanation was added in chapters 3 and 4. Thank you for your invaluable comments to improve paper logic.
Modification: Chapters 3 and 4

Round 2

Reviewer 4 Report

I have read the new version of the paper and I believe that the authors have improved the link between the different investigations. The authors' responses to my comments are satisfactory, so I think the paper is worthy of publication.